# CONDITIONING SEQUENCE-TO-SEQUENCE MODELS WITH LEARNED ACTIVATION FUNCTIONS

**Alberto Gil C. P. Ramos**[1,*], **Abhinav Mehrotra**[1,*], **Nicholas D. Lane**[1,2], **Sourav Bhattacharya**[1]

[1]Samsung AI Centre, Cambridge, UK [2]University of Cambridge, UK

{a.gilramos,a.mehrotra1,nic.lane,sourav.b1}@samsung.com

## ABSTRACT

Conditional neural networks play an important role in a number of sequence-to-sequence modeling tasks, including personalized sound enhancement (PSE), speaker dependent automatic speech recognition (ASR), and generative modeling such as text-to-speech synthesis. In conditional neural networks, the output of a model is often influenced by a conditioning vector, in addition to the input. Common approaches of conditioning include input concatenation or modulation with the conditioning vector, which comes at a cost of increased model size. In this work, we introduce a novel approach of neural network conditioning by learning intermediate layer activations based on the conditioning vector. We systematically explore and show that learned activation functions can produce conditional models with comparable or better quality, while decreasing model sizes, thus making them ideal candidates for resource-efficient on-device deployment. As exemplary target use-cases we consider (i) the task of PSE as a pre-processing technique for improving telephony or pre-trained ASR performance under noise, and (ii) personalized ASR in single speaker scenarios. We find that conditioning via activation function learning is an effective modeling strategy, suggesting a broad applicability of the proposed technique across a number of application domains.

## 1 INTRODUCTION

The use of latent or embedding vectors to condition neural networks is common in a number of machine learning tasks. For example, in personalized sound enhancement (PSE) prior knowledge of a speaker's voice profile is used to filter out noise from an audio recording (Choi et al., 2021; Hu et al., 2020; Wang et al., 2020; 2019), in speaker-dependent or personalized Automatic Speech Recognition (ASR) a speaker's voice profile is also used to improve the overall transcription quality (Bell et al., 2021; He et al., 2019; Samarakoon et al., 2016; Saon et al., 2013), in personalized Text-To-Speech (TTS) systems synthesized voice profile can be purposefully adapted (van den Oord et al., 2018), in computer vision tasks, such as in visual reasoning applications (Chen et al., 2019; Perez et al., 2018), questions are used as the conditioning vectors, and similarly in conditional domain adaptation techniques (Long et al., 2018). More generally, conditional neural networks accept an additional conditioning vector, that influences how the networks process inputs to produce outputs.

Most common approaches for conditioning neural networks either concatenate or modulate the conditioning vector with one or more intermediate representations of the network, before the updated representations are processed by a layer, e.g., see Figure 1a-b for an overview. Such conditioning techniques come at the cost of significantly increasing the network parameters by inflating the dimensionality of intermediate features or by requiring additional network parameters proportional to the dimensionality of the conditioning vectors and the number of channels present in the representations.

In this work, we introduce a new approach for neural network conditioning and show that effective conditioning can be achieved by learning layer-specific activation functions based on the conditioning vectors. Unlike previous approaches, our *learned activation* (LA) functions approach does not increase the model parameters significantly, since the number of additional parameters it requires depends only on the size of the conditioning vector, and not on the shape of the feature representations nor on the output shapes of any subsequent layers. LA uses the conditioning vectors to dynamically modify intermediate activation functions, which are $\mathbb{R} \to \mathbb{R}$ (1-D) mappings, see Figure 1c. Dynamic

---

*Equal contribution

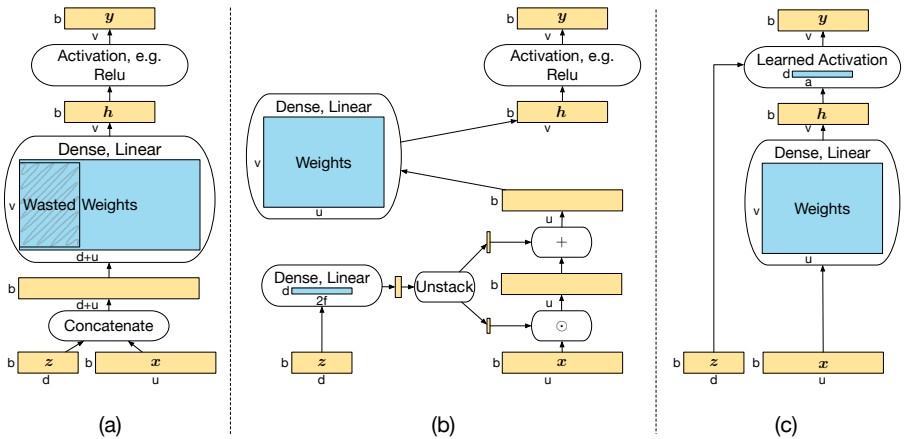

Figure 1: Overview of neural network conditioning methods with two inputs, $\boldsymbol{z} \in \mathbb{R}^{b \times d}$ conditioning vector and $\boldsymbol{x} \in \mathbb{R}^{b \times \cdots}$ input feature: (a) conditioning via concatenation, (b) modulation and (c) the proposed learned activation (LA) functions, illustrated here together with a dense layer.

LAs allow us to extract important insights on how the model is being adapted for different conditioning vectors by simply analyzing the learned activation functions, which are agnostic to the input.

In this work we consider two popular and timely sequence-to-sequence modeling tasks: *personalized sound enhancement* (PSE) and *personalized ASR* (PASR) as exemplary tasks for modeling conditional neural networks. PSE leverages a speaker embedding vector, i.e., a unique encoded voice profile of a speaker, for filtering out everything but the speaker's voice from an audio recording corrupted with babble or ambient noise (Wang et al., 2020; 2019). Speaker embeddings can generally be obtained by using an embedding network once during an enrolment phase, e.g., from recorded utterances like "Hey Alexa" or "Hey Bixby". PSE networks can be evaluated for improving signal quality for telephony or can also be used to improve the quality of downstream tasks, such as a pre-trained ASR. PASR on the other hand, aims to improve ASR quality in single speaker scenarios, by leveraging user's enrollment data to enhance performance directly without any pre-processing.

We evaluate LA on three PSE model families (§4.2), on two datasets Librispeech (Panayotov et al., 2015) and Voxforge (vox, 2006). We show that the use of LA on all models achieves competitive performance to conditioning using *concatenation* and *modulation* approaches. At the same time, concatenation approaches require 10–95% more model parameters over our solution. For PASR, we compare the performance of non-personalized and personalized versions of the conformer encoder model architecture (Gulati et al., 2020) with a dense layer for character or subword text encoder trained with CTC loss and decoded via CTC decoder (§5.2). Results on Librispeech (Panayotov et al., 2015) show that LA yields effective ASR personalization, i.e., increased performance across all users while using two seconds of enrollment data, with up to 19% relative WER improvement across all users, and up to 40% for some users.

## 2 RELATED WORK

**Conditioning through tensor concatenation**   The concatenation of intermediate features with latent vectors before being passed to subsequent layers is the most common form of conditioning, but leads to inflated model sizes, which hinder their deployment to low-resource constrained devices (Wang et al., 2019; 2020; Valin et al., 2020; Hu et al., 2020; Comon & Jutten, 2010; Wang et al., 2021; Lam et al., 2021; Vincent et al., 2018; Luo & Mesgarani, 2019; Xu et al., 2020; Abdulatif et al., 2020; Wisdom et al., 2020; Isik et al., 2020; Luo et al., 2020; Choi et al., 2021; Snyder et al., 2018; Panayotov et al., 2015; Bell et al., 2021; He et al., 2019; van den Oord et al., 2018). Our approach overcomes this issue and learns smaller models with competitive quality.

**Conditioning via normalization layers**   Another family of techniques is based on having per channel learned linear transformation based on the conditioning vector, which yields a form of conditional normalization (Perez et al., 2018; de Vries et al., 2017; Huang & Belongie, 2017). Our approach distinguishes itself from this line of work since it learns the non-linear components of the network rather than the linear ones, and thus offers greater modelling at lower cost.

**Learning activations without conditioning**   Outside the context of conditional neural networks, there is work on learning an optimal activation function as a good default to be used in all layers of

most neural network architectures (Apicella et al., 2021; Liu et al., 2020; Ramachandran et al., 2018; Agostinelli et al., 2015). Our work instead i) learns conditional activations based on conditioning vectors, and ii) uses differently learnt activations throughout the network.

## 3  CONDITIONING BY LEARNING ACTIVATIONS

Conditioning a neural network corresponds to using some form of latent vector or additional prior knowledge, e.g., encoded typically in the form of an embedding vector, to influence the output of the network for a given input. We show in the evaluation that the influence of the conditioning vector can be successfully modeled by learning a weighted combination of a set of basic activation functions (§3.1) and carefully selecting hyperparameters (§3.2).

### 3.1  DYNAMICALLY LEARNED ACTIVATIONS

To introduce our conditioning approach using dynamically weighted activation functions, we begin by defining a few key variables: $\boldsymbol{z} \in \mathbb{R}^{b \times d}$ is a 2-D tensor representing $b$ number of $d$-dimensional conditioning vectors, where $b$ is the batch size, $\boldsymbol{x} \in \mathbb{R}^{b \times \cdots}$ is the input data with arbitrary tensor dimensions (e.g., vectors, matrices, higher order tensors), $\boldsymbol{h} \in \mathbb{R}^{b \times \cdots}$ is the hidden pre-activation of the previous layer, and $\{A_i : \mathbb{R} \to \mathbb{R}\}_{i=1}^a$ is an ordered family of $a$ basic activations, e.g., $\{\mathrm{relu}, \mathrm{sigmoid}, \mathrm{tanh}\}$ (we will discuss below more on the elements of this ordered set of activations). Dynamically learned activation (LA) functions require a small number of trainable variables $\boldsymbol{w} \in \mathbb{R}^{d \times a}$ and $\boldsymbol{b} \in \mathbb{R}^{1 \times a}$, and are defined by transforming input tuples $(\boldsymbol{z}, \boldsymbol{h})$ into output tensors $\boldsymbol{y}$ through (note the parallel between the following equations and the pictorial representation in Figure 1c, where there is a direct correspondence on the roles of $\boldsymbol{z}$, $\boldsymbol{h}$ and $\boldsymbol{y}$ between the formulas and the diagram):

$$\boldsymbol{s} := \mathrm{softmax}_{\mathrm{rowwise}}(\boldsymbol{z} \times \boldsymbol{w} + \boldsymbol{b}),$$

$$h \in \mathbb{R} \mapsto \mathrm{LA}(h \,|\, \boldsymbol{z}_j) := \sum_{i=1}^a \boldsymbol{s}_{j,i} \cdot A_i(h),$$

$$\boldsymbol{y}_{j,\ldots} := \mathrm{LA}_{\mathrm{elementwise}}(\boldsymbol{h}_{j,\ldots} \,|\, \boldsymbol{z}_j),$$

where $\mathrm{softmax}_{\mathrm{rowwise}}$ denotes the matrix-matrix operation with entries defined rowwise via $\mathrm{softmax}_{\mathrm{rowwise}}(\boldsymbol{m})_j := \mathrm{softmax}(\boldsymbol{m}_j)$, and $\mathrm{LA}_{\mathrm{elementwise}}(\boldsymbol{h}_{j,\ldots} \,|\, \boldsymbol{z}_j)$ denotes the result of applying $\mathrm{LA}(\cdot \,|\, \boldsymbol{z}_j)$ elementwise to each entry of $\boldsymbol{h}_{j,\ldots}$.

Note that during training, batch size is greater than one in which case $\boldsymbol{z}$ is a matrix, whereas during inference batch size is one for each user in which case $\boldsymbol{z}$ is a row vector. So during training $\boldsymbol{z}_j$ is the conditioning vector corresponding to the $j$-th sample, but during inference $\boldsymbol{z}_j$ is the conditioning vector corresponding to the $j$-th user in the discussion below.

Figure 1 highlights the qualitative and quantitative differences between our approach (Figure 1c) and existing state-of-the-art (SOTA) solutions (Figure 1a-b). *Concatenation approaches* as depicted in Fig. 1a can be defined by $A_{\mathrm{elementwise}}(F_{\boldsymbol{\theta}}([\boldsymbol{z}_j^\top, \boldsymbol{x}_{j,\ldots:}^\top]^\top))$, where $F_{\boldsymbol{\theta}}$ denotes a common learnable transformation such as a dense, conv1d, conv2d or LSTM layer without activation and $A : \mathbb{R} \to \mathbb{R}$ denotes a generic activation such as the rectified linear unit. Whereas, as shown in Fig. 1b, *modulation approaches* are defined as $A_{\mathrm{elementwise}}(G_{\boldsymbol{\eta}}(U_{\boldsymbol{\mu}}(\boldsymbol{z}_j) \odot \boldsymbol{x}_{j,\ldots:} + V_{\boldsymbol{\nu}}(\boldsymbol{z}_j)))$, where $G_{\boldsymbol{\eta}}$ represents a common learnable transformation (which may be slightly different than $F_{\boldsymbol{\theta}}$ given their inputs have different shapes) and $U_{\boldsymbol{\mu}} : \mathbb{R}^d \to \mathbb{R}^f$ and $V_{\boldsymbol{\nu}} : \mathbb{R}^d \to \mathbb{R}^f$ are often taken as linear mappings (where $d$ denotes the conditioning vectors dimension and $f$ denotes the number of channels of $\boldsymbol{x}$).

In comparison to these SOTA approaches, our learned activation functions mechanism can simply be formulated as $\mathrm{LA}_{\mathrm{elementwise}}(G_{\boldsymbol{\eta}}(\boldsymbol{x}_{j,\ldots}) \,|\, \boldsymbol{z}_j)$. For simplicity, we use a dense layer in the figure (i.e., we take $F_{\boldsymbol{\theta}}$ and $G_{\boldsymbol{\eta}}$ to be dense layers and $f = 1$), however, our approach is general and can be applied to any layer type, e.g, dense, conv1d, conv2d and LSTM. From a qualitative point of view, LAs conditions the non-linearities rather than the linearities of the network. From a quantitative point of view, Figure 1 also highlights the superior compression abilities achieved by our proposed method.

In particular, concatenation approaches (Figure 1a) require $(d + u) * v$ parameters and modulation approaches (Figure 1b) require $u * v + d * 2 * f$, where $f$ denotes the number of channels in the input tensor $\boldsymbol{x}$, which per channel is multiplied by a scaling factor and offset by a bias term. The learned activation functions approach on the other hand, requires only $u * v + d * a$ parameters. For example, in most audio and video applications, our approach yields significant savings of $d * (v - a)$ given that $d \gg 1$ and $v \gg a$, when compared to concatenation approaches. The saving becomes $d * (2 * f - a)$ when compared to modulation approaches, given that often in such cases $2 * f - a > 0$.

For temporal data, such as audio and video, the savings of our approach are even more significant as in those applications the conditioning vector is concatenated with each time frame of the inputs and for $f \gg 1$ the model requires larger parameter size. In our approach the operational requirements do not change across data type, i.e., tensor dimensions.

## 3.2 HYPER-PARAMETERS

**Basic Activations** As discussed above, our conditioning approach relies on a set of standard activations provided for modelling. Therefore, the selection of these basic activations is an important hyper-parameter in our approach. Following are the key characteristics that should be taken into account when searching for an optimal set of activations:

- Dynamically learned activation coefficients $s_j$ are all positive and sum to one. Thus, some families of activations will generate learned activations that remain in the same family, however, this not always true. For example, if all basic activations are all increasing functions, so is the learned activation. Monotonic increasing property of the learned activation may not be retained if we include both increasing and decreasing functions in the basic activation set.
- The fact that $s_j$ are in the probability simplex means that the ranges of the basic activations play an important role in the final learned activations. One could consider basic activations with i) unbounded, ii) upper bounded, iii) lower bounded or iv) bounded ranges.
- Finally, the basic activations could instead be picked from well known basis functions. For instance, Fourier basis or Chebyshev polynomials could be used to create bounded learned activations with compact support, and Hermite polynomials to create unbounded learned activations on the real line. Note that this direction of investigation is outside the scope of our paper.

**Initialization Choices** Another important hyper-parameter to consider is what are the values of the dynamically learned coefficients $s_j$ at initialization time for most conditioning vectors $z_j$. Given $s_j$ is a probability distribution, to start training with an unbiased model, they could be at initialization time approximately uniform, i.e., providing roughly the same weight to each of the basic activations for most $z_j$. Alternatively, they could be such that for most $z_j$ at the beginning of training the learned activations have a desired shape. Therefore, trainable variables $w$ and $b$ should be initialized considering i) the statistics of the conditioning vectors $z_j$, and ii) the shapes of the basic activations.

## 4 EXPERIMENTS PART I: PSE FOR TELEPHONY AND PRE-TRAINED ASR

In this section we consider the task of PSE, with the aim of enhancing a user's voice to improve telephony and pre-trained ASR quality under babble (i.e., other speaker's voices) or ambient noise, in real-world scenarios with only two seconds of enrolment audio of non-constrained speech.

### 4.1 DATASETS

We consider two representative datasets: LibriSpeech (Panayotov et al., 2015) and Voxforge (vox, 2006), which cover English and Spanish respectively. For the English training set, we take 100h and 360h of clean speech from LibriSpeech, which are recorded using close-talk microphones, without any background noise. Although, supervised learning approaches have shown great performance in sound enhancement tasks, it is often impossible to collect the mixed audio and its ground-truth sound sources from the same audio environment (Wisdom et al., 2020). Furthermore, to evaluate performances of learned activation on realistic data, we create a training dataset by taking 32.5h of Spanish audio from Voxforge, which are mainly recorded with a far-field microphone and contain noise. Both datasets have audio recorded at 16kHz.

The raw datasets consist of tuples (speech, transcript, speaker identifier). For each pair of different speakers $a$ and $b$, we collect two speech samples from $a$ and one from $b$ to create 4-tuples $(\text{speech}_a, \text{speech}'_a + \text{speech}_b, \text{speech}'_a, \text{text}'_a)$, with elements being: $i)$ a reference speech from user $a$ to be passed through a speaker embedding model (Snyder et al., 2018) to create the conditioning vector, $ii)$ a speech from user $a$, different from that given to the embedding model, which is corrupted (at various db levels) with speech from user $b$, $iii)$ the ground-truth speech by user $a$ present in the corrupted speech, and, $iv)$ a transcript of the ground-truth speech by user $a$. All PSE models are trained in a supervised fashion, where the goal is to recover $\text{speech}'_a$ from $\text{speech}'_a + \text{speech}_b$ as closely as possible for telephony applications and such that $\text{ASR}(\text{PSE}(\text{speech}'_a + \text{speech}_b)) = \text{text}'_a$ for ASR applications. We also evaluate performances under ambient acoustic noise by constructing $(\text{speech}_a, \text{speech}'_a + \text{noise}, \text{speech}'_a, \text{text}'_a)$ tuples, where noise is sampled from the DEMAND dataset (Thiemann et al., 2013).

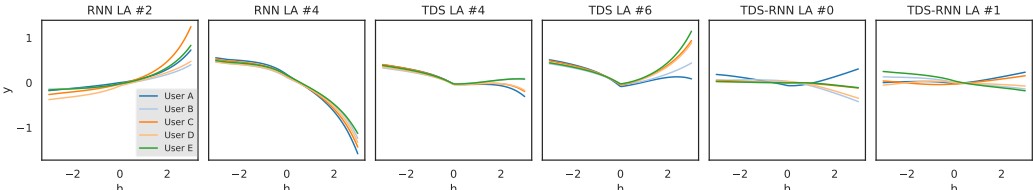

Figure 2: Learned activations (LA) for models trained on Voxforge, for speaker embedding vectors of randomly selected users in a PSE application. To take into account that basic activations may have different ranges, the plots show LAs minus average of basic activations, i.e., $\mathrm{LA}(h\,|\,\boldsymbol{z}_j) - a^{-1}\sum_{i=1}^{a} \mathrm{A}_i(h)$ for $h \in [-3,3]$ and values of $j$ corresponding to the selected users. The plots highlight that LA exhibit different profiles across different users and layers.

Table 1: Source-to-Distortion Ratio improvement (SDRi) (higher the better) and Word-Error-Rate (WER) (lower the better) performances on LibriSpeech (English) for models with conditioning via learned activations (LAs), modulation (FiLM) and concatenation, across models with different number of conditioning layers (#Cond.).

| Model | Type | #Cond. | Params (M) | SDRi | | WER | | |
| --- | --- | --- | --- | --- | --- | --- | --- | --- |
| | | | | Babble | Ambient | Clean | Babble | Ambient |
| Identity | - | - | - | - | - | 8.2 | 89.4 | 17.3 |
| RNN | LA | 8 | 4.31 | 8.47 | 12.02 | 8.3 | 24.8 | 15.2 |
| RNN | FiLM | - | 5.34 | 7.39 | 11.41 | 9.4 | 31.2 | 16.4 |
| RNN | Concat. | - | 6.13 | 8.57 | 12.19 | 8.3 | 24.5 | 15.0 |
| TDS | LA | 42 | 2.51 | 7.33 | 11.58 | 8.2 | 33.0 | 15.7 |
| TDS | FiLM | - | 4.10 | 6.74 | 11.34 | 8.3 | 37.5 | 16.9 |
| TDS | Concat. | - | 4.89 | 7.78 | 11.72 | 8.3 | 30.0 | 15.7 |
| TDS–RNN | LA | 2 | 9.56 | 8.96 | 12.44 | 8.3 | 21.9 | 14.3 |
| TDS–RNN | FiLM | - | 9.87 | 8.36 | 12.08 | 9.0 | 25.4 | 15.2 |
| TDS–RNN | Concat. | - | 10.48 | 9.49 | 12.49 | 9.3 | 18.2 | 14.3 |

Table 2: Source-to-Distortion Ratio improvement (SDRi) (higher the better) and Word-Error-Rate (WER) (lower the better) performances on VoxForge (Spanish) for models with conditioning via learned activations (LAs), modulation (FiLM) and concatenation, across models with different number of conditioning layers (#Cond.).

| Model | Type | #Cond. | Params (M) | SDRi | | WER | | |
| --- | --- | --- | --- | --- | --- | --- | --- | --- |
| | | | | Babble | Ambient | Clean | Babble | Ambient |
| Identity | - | - | - | - | - | 1.2 | 89.1 | 9.2 |
| RNN | LA | 8 | 4.31 | 6.96 | 9.81 | 1.9 | 20.8 | 6.7 |
| RNN | FiLM | - | 5.34 | 5.92 | 8.33 | 3.0 | 25.5 | 8.2 |
| RNN | Concat. | - | 6.13 | 6.84 | 9.64 | 1.4 | 22.8 | 6.4 |
| TDS | LA | 42 | 2.51 | 5.37 | 8.21 | 1.4 | 33.3 | 7.7 |
| TDS | FiLM | - | 4.10 | 6.09 | 8.45 | 1.8 | 24.3 | 7.3 |
| TDS | Concat. | - | 4.89 | 5.19 | 8.02 | 1.6 | 35.6 | 7.6 |
| TDS–RNN | LA | 2 | 9.56 | 7.17 | 10.36 | 1.8 | 19.5 | 6.0 |
| TDS–RNN | FiLM | - | 9.87 | 7.23 | 9.91 | 2.3 | 18.0 | 6.4 |
| TDS–RNN | Concat. | - | 10.48 | 7.69 | 10.50 | 1.9 | 15.2 | 5.9 |

## 4.2 BASELINE MODELS

A variety of architectures have been proposed lately for speech enhancement. In this work, we consider architectures suitable for low-latency on-device deployment that have small memory footprints and operate in real-time. With that in mind, we use the following three architectures as baselines:

- **RNN.** These are RNN models with skip-connections (Takeuchi et al., 2020; Wang et al., 2020) (c.f., Figure 5a in Appendix A).
- **TDS.** These models (Figure 5b in Appendix A) comprise of stacked time-depth-separable (TDS) convolution blocks (Isik et al., 2020; Luo & Mesgarani, 2019).

- **TDS-RNN.** These are encoder-decoder style models with a separator block in between (Figure 5c in Appendix A). Encoder and decoder are made of stacked TDS blocks and the separator of stacked RNN blocks (Choi et al., 2021).

Furthermore, we consider a different number of conditioning layers (#Cond.) for each model family. In particular, for RNN 8, for TDS 21 and for TDS–RNN 2 conditioning layers leading to 8, 42 and 2 nodes in the models being influenced by the conditioning information (c.f., Figure 5 in Appendix A). We have done so to investigate also how the number of conditioning layers influences the personalization effects from the conditioning vectors (c.f., Appendix A.2).

In our experiments we apply LAs in all three families of models and perform extensive comparisons with the baselines in terms of performance, model size and latency. In particular, we take the best performing baseline concatenation architectures, not the weights, and for every layer with a conditioning concatenation, we replace that with a modulation layer or a conditionally learned activation. Finally we train the modulation models and the conditionally learned activation models from scratch, without performing any architectural hyper-parameter optimization (HPO). As a representative of light weight-modulation approaches, we consider the Feature-wise Linear Modulation (FiLM) (Perez et al., 2018) mechanism throughout. Notably, our evaluation experiments give more advantages to the baseline concatenation models.

### 4.3 HYPER-PARAMETERS

**Basic Activations**    The choice of basic activation functions is an important hyper-parameter in our approach and we consider a comprehensive list of popular activation functions used in deep learning. Specifically, we use the following ordered set as the basic activation functions: $\{A_i : \mathbb{R} \to \mathbb{R}\}_{i=1}^a = \{$elu, exponential, hard_sigmoid, linear, relu, selu, sigmoid, softplus, softsign, swish, tanh$\}$.

**Initialization Choices**    We consider enrolment data (e.g., a short pre-defined utterance spoken by the user once) to extract (using a $\ell_2$-normalized X-Vector model (Snyder et al., 2018)) a speech embedding vector of the target user, i.e., the owner of the device. This speech embedding is used as the conditioning vector in PSE models to enhance the user's voice and suppress unwanted voices and background noise. Although, the embedding vectors in the training set are not perfectly zero mean and unit variance, the fact that they are $\ell_2$-normalized motivates us to take for initialization of the weight matrix in the LAs the Glorot uniform initializer and for the bias vector the zero initializer.

### 4.4 TRAINING SETTINGS

All models considered in this work operate in the time-frequency domain. We apply Short Time Fourier Transform (STFT) to extract 512 coefficients using a 32 ms window and with a stride of 16 ms. Models are trained in a data-parallel way with four GPUs using TensorFlow and Horovod. We use batch size of 64, learning rates in $[1e{-}5, 1e{-}3]$, exponential decay learning rate scheduler and early stopping (max epochs 100).

### 4.5 EVALUATION METRICS

We evaluate our models in terms of source-to-distortion ratio improvement (SDRi) (Comon & Jutten, 2010) and word-error-rate (WER) metrics. A high SDRi indicates the improvement in signal quality for telephony and a low WER quantifies good transcription quality of the downstream ASR task. We use open-source Silero (sil, 2021) ASR models (English and Spanish) in our evaluations.

### 4.6 RESULTS

This subsection presents evaluation results for quantifying the effectiveness of learned activations (LA), compared to concatenation- and modulation-based conditioning approaches, on LibriSpeech (Table 1) and Voxforge (Table 2) datasets. The results show that models with all three conditioning approaches improve telephony (high SDRi) and ASR (low WER) performances on both datasets. We observe that LA and concatenation-based conditioning approaches yield very similar performances, indicating high efficacy of the proposed LA approach. Note that when evaluating ASR, it is also important that a good PSE model shows almost no degradation in WER, when clean audio is presented. Interestingly in some cases in the concatenation-based approach, we see that higher success in removing babble noise, comes with an adverse impact in clean speech WER, mainly due to over-suppression (Wang et al., 2020). Furthermore, Table 1–2 show that both concatenation and modulations produce significantly larger models, i.e., up to 95% larger, compared to LA approaches.

Tables 1–2 highlight the trade-off between the number of parameters and quality metrics. However, it is important also to consider the impact of the different approaches in terms of FLOPS and latency,

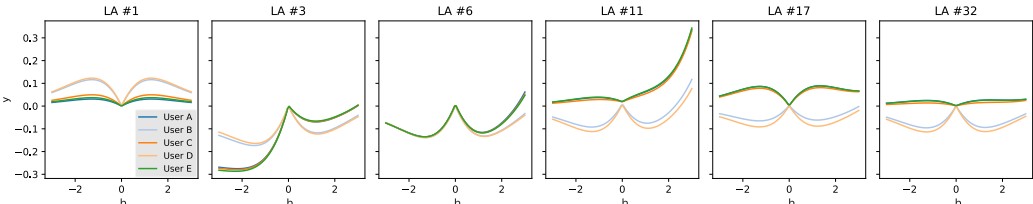

Figure 3: Learned activations (LA) for models trained on LibriSpeech, for speaker embedding vectors of randomly selected users in a PASR application. To take into account that the non-personalized basic activations are relu or swish, the plots show LAs minus average of those two activations, i.e., $\text{LA}(h \mid \boldsymbol{z}_j) - (\text{relu}(h) + \text{swish}(h))/2$ for $h \in [-3, 3]$ and values of $j$ corresponding to the selected users. The plots highlight that LA exhibit distinct profiles across different users and layers.

Table 3: FLOPs and latency (for one second audio) for PSE models with conditioning via learned activations (LAs), modulation (FiLM) and concatenation (Concat.). Note FLOPs may not be completely accurate given the common lack of support of profiling tools for non-standard layers or operations.

| Model | Type | FLOPs (M) | Latency CPU (ms) | Latency GPU (ms) |
|---|---|---|---|---|
| RNN | LA | 1254.85 | 462.997 ($\pm$14.719) | 39.156 ($\pm$01.339) |
| RNN | FiLM | 1254.85 | 465.785 ($\pm$14.612) | 35.769 ($\pm$01.344) |
| RNN | Concat. | 1802.60 | 537.766 ($\pm$11.403) | 46.716 ($\pm$00.903) |
| TDS | LA | 1507.43 | 127.420 ($\pm$02.996) | 78.998 ($\pm$01.963) |
| TDS | FiLM | 1507.43 | 95.535 ($\pm$04.861) | 70.265 ($\pm$01.975) |
| TDS | Concat. | 2906.46 | 123.580 ($\pm$03.440) | 84.614 ($\pm$02.479) |
| TDS–RNN | LA | 4406.60 | 380.014 ($\pm$16.604) | 233.748 ($\pm$05.315) |
| TDS–RNN | FiLM | 4406.60 | 387.325 ($\pm$22.101) | 238.787 ($\pm$11.882) |
| TDS–RNN | Concat. | 4543.92 | 394.809 ($\pm$20.247) | 237.437 ($\pm$06.928) |

as shown in Table 3 (where latency was measured by passing one second of data one hundred times by the models and summarized via mean and standard deviation), which demonstrates LAs are competitive in this dimension. Specifically, LA is faster than standard concatenation approach with comparable quality, and slower than FiLM but with higher quality (i.e., higher SDRi and lower WER).

To gain a better understanding of how LAs adapt based on conditioning vectors, Figure 2 presents plots centered at the origin of the LAs for a PSE application conditioned on speaker embedding vectors extracted from enrolment audio from randomly selected users. In particular, we can see that for all models i) in each layer, different users have different LAs, and ii) each user has different LAs across different layers. As such the visualization of the internal mechanisms of LAs, provides a tangible way to improve how we understand the manner in which models leverage conditioning vectors to influence predictions.

## 5 EXPERIMENTS PART II: PERSONALIZED ASR

In this section we consider the problem of personalized ASR, with the objective of improving ASR performance in single speaker scenarios. This is a challenging task, especially in realistic setups with only a couple of seconds of enrolment audio with non-constrained speech which we explore below.

### 5.1 DATASETS

We consider LibriSpeech (Panayotov et al., 2015) 1000h of English annotated audios with transcripts. We augment each sample tuple (audio, transcript) to include two seconds enrollment audio from the same user (audio, transcript, enrollment). We ensure the enrollment audio is taken from a different sample from the same user. For diversity during training the enrollment audio is selected randomly under the aforementioned constraint, whereas for evaluation it is always the same.

### 5.2 BASELINE MODELS

**ASR.** For non-personalized ASR models we take the conformer encoder model architecture (Gulati et al., 2020) with a dense layer for character (with 28 characters, namely: 'A'–'Z', ' ' and '") or subword (with 753 subwords) text encoder trained with CTC loss and decoded via CTC decoder. These models take as input an audio and output a predicted transcript.

**PASR.** For personalized ASR we take the ASR architecture but replace the ReLU and Swish activations with our proposed LAs (see §3.1). We consider basic activations (see §3.2) containing ReLU and Swish, and an initialization of the bias (see, §3.2) so that at initialization time, the LAs

replacing ReLU are approximately ReLU, and the LAs replacing Swish are approximately Swish, a profile which changes during training as a mechanism to enable personalization. As a similar mechanism to the bias offset in LAs, we altered FiLM's definition $A_{\text{elementwise}}(G_{\boldsymbol{\eta}}(U_{\boldsymbol{\mu}}(\boldsymbol{z}_j) \odot \boldsymbol{x}_{j,...,:} + V_{\boldsymbol{\nu}}(\boldsymbol{z}_j)))$ to $A_{\text{elementwise}}(G_{\boldsymbol{\eta}}((1 + \alpha \cdot \tanh(U_{\boldsymbol{\mu}}(\boldsymbol{z}_j))) \odot \boldsymbol{x}_{j,...,:} + \beta \cdot \tanh(V_{\boldsymbol{\nu}}(\boldsymbol{z}_j))))$ with $\alpha$ and $\beta$ as trainable variables, so that at initialization FiLM is approximately the identity function for most conditioning vectors. As with non-personalized ASR, we consider character and subword text encoders trained with CTC loss and decoded by CTC decoder. These models take as input an audio and a fixed speaker embedding vector for each user computed from enrollment data by a speaker embedding network (Snyder et al., 2018), and output a predicted transcript.

## 5.3 HYPER-PARAMETERS

For consistency with previous experiments, we take as the basic activations those used in §4.3. For initialization choices as mentioned earlier we introduce a bias offset so that at initialization the softmax $\boldsymbol{s}$ described in §3.1 is approximately one-hot to leverage the inductive bias of activations proven to work well in non-personalized settings. Interestingly, we observed in our experiments that subword models seem to be more sensitive to this hyper-parameter than character models.

## 5.4 EVALUATION METRICS

For character and subword models we consider greedy (beam search with tops paths and beam width equal to one) and beam search (with a beam width of four) CTC decoders. Once predicted transcripts are produced from the models, they are compared with ground-truth transcripts in terms of word error rate (WER) and character error rate (CER).

## 5.5 RESULTS

We consider the performance of PASR models created from existing ASR models by replacing their fixed activations with LAs. To delve deeper into the effect of LAs on PASR, we consider two dimensions to our experiments, namely, in terms of whether i) training is performed only on the speaker embedding model as well as on the weights and biases of LA, or ii) training is done on all parameters. In both cases, optimization is done with respect to lower CTC loss. These settings are designed on the one hand to understand what are the effects of taking pre-trained ASR models and personalizing them by replacing their activations with LAs, and on the other hand to understand the capacity and convergence properties that LAs provide for PASR. Before a fine-grain analysis of the results below, note that similarly to Figure 2 for PSE, Figure 3 shows that LAs consistently create different activation profiles across users and layers for PASR as well.

### 5.5.1 TRAIN SELECTED PARAMETERS

Leaving the parameters of the original ASR intact, allows for switching between the original static activations and the LAs, which might be beneficial for deployment scenarios lacking clean enrolment data. We investigate this scenario by fine-tuning only the speaker embedding model and LA parameters, which results in an overall 2% WER relative improvement which is a modest but significant improvement given an already low WER/CER of the original ASR. Figure 4 (top) shows that for some users this improvement is even more significant, as high as 15% WER relative improvement.

### 5.5.2 TRAIN ALL PARAMETERS

In this section we investigate the performance of PASR models derived from training all their parameters after initializing their non-personalized components from existing ASR models.

**Results for all users together** Table 4 contains WER/CER results aggregated across all users of LibriSpeech test-clean. Overall the results show a consistent improvement in performance, i.e., lower WER/CER for personalized ASR via LAs compared with non-personalized ASR. The results also show that subword models yield better WER, but character models give better CER, with the gains through personalization being higher for character than subword models, being as high as 19% relative WER improvement across all users.

**Results for each user separately** Figure 4 (bottom) shows plots of WER relative (in terms of percentage) and absolute improvement per each user of LibriSpeech test-clean for character models with CTC greedy and beam search decoders. Each plot shows three curves, namely: i) the horizontal blue line $y = 0$ representing the performance of the non-personalized ASR, ii) the orange curve representing the performance of the non-personalized ASR with its activations replaced by randomly initialized LAs, and iii) the green curve representing the performance of this personalized ASR fine-tuning speaker embedding model, weights and biases of LAs, as well as the parameters of the non-personalized ASR. The results show an initial mixed degradation/improvement per user when

Table 4: WER and LER results on LibriSpeech test-clean for ASR and PASR (via FiLM modulation and proposed LA functions approaches) with two seconds of enrollment data, for character and subword text encoders and CTC greedy (G) and beam search (BS) with a beam width four decoders.

| Models | ASR | | | | PASR | | | | | | | |
|---|---|---|---|---|---|---|---|---|---|---|---|---|
| Type | - | | | | FiLM | | | | LA | | | |
| Encoder | Char | | Subword | | Char | | Subword | | Char | | Subword | |
| Decoder | G | BS | G | BS | G | BS | G | BS | G | BS | G | BS |
| WER | 8.93 | 8.77 | 7.69 | 7.62 | 7.41 | 7.27 | 6.99 | 6.93 | 7.26 | 7.15 | 6.41 | 6.35 |
| CER | 2.65 | 2.62 | 5.04 | 5.01 | 2.20 | 2.18 | 4.78 | 4.75 | 2.17 | 2.14 | 4.39 | 4.36 |

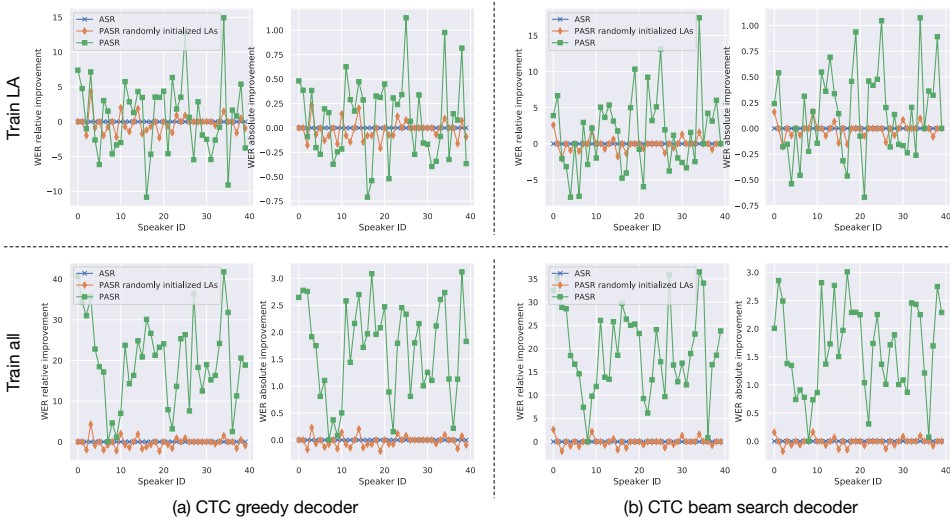

(a) CTC greedy decoder   (b) CTC beam search decoder

Figure 4: WER relative and absolute improvement on LibriSpeech test-clean fine-tuning speaker embedding and LAs (top), training everything (bottom) for CTC greedy (a) and beam search with beam width four (b) decoders, with two seconds of enrollment data, for character text encoder.

replacing the original activations with LAs as expected without training (ensured by an appropriate choice of the bias offset hyper-parameter as described in §5.3), and the benefits of training the personalized ASR, with significant gains in WER per user, in terms of relative and absolute WER. Furthermore, the gains can be seen to reach up to 40% and 35% relative improvement for some users for CTC greedy decoder and CTC beam search decoder respectively, which is a significant improvement in the domain of ASR.

## 6 LIMITATIONS

Our evaluations are limited to two supervised audio domains, namely: i) PSE to improve telephony and pre-trained ASR quality for English and Spanish, under babble and ambient noise, and ii) personalized ASR for single speaker scenarios over English, based only on two seconds of enrollment data. We nevertheless see value in assessing our approach in generative or image domains. Another limitation stems from the fact that the evaluation is performed with speaker embeddings computed from clean enrolment data, which might not be available at deployment time. So the robustness to this source of issues is not explored in our experimental setting. This limitation is common in most, if not all studies, in these application domains.

## 7 CONCLUSIONS

We introduced a new general method for designing conditional neural networks. Unlike state-of-the-art techniques, which are based on concatenation or modulation strategies, we presented a new form of conditioning by dynamically learning activations. By conducting an extensive set of experiments, we highlighted the efficacy of our modeling technique in terms of quality metrics and model sizes. Although, we mainly focus on PSE under babble and ambient scenarios and personalized ASR for single speaker cases as the main application domains, we believe our approach will open ways for the generalized use of dynamically learned activation functions in many application domains.

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

# A APPENDIX

## A.1 BASELINE MODEL FOR PSE

Figure 5 presents a pictorial representation of the PSE models introduced in §4.2.

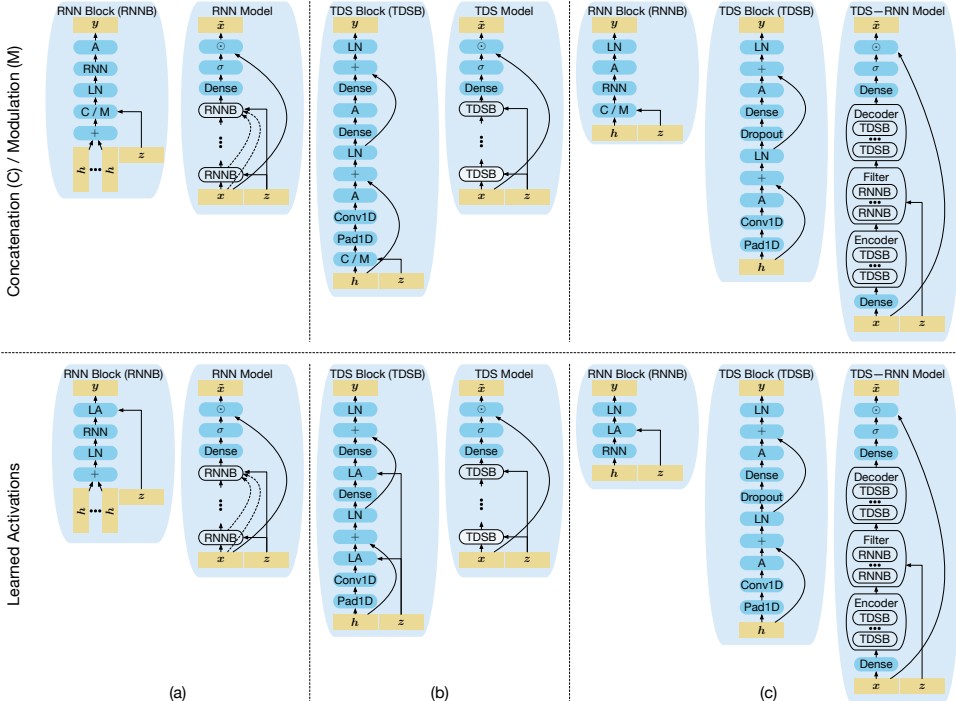

Figure 5: A pictorial representation of the RNN (a), TDS (b) and TDS–RNN (c) models used in the evaluation of conditioning based on concatenation (top, C), modulation (top, M) and based on the proposed learned activations (LAs) (bottom).

## A.2 REGULARIZERS

We also investigate the use of regularizers during training for learning activation functions, in order to boost deployment and conditioning efficiency. We consider two regularizers: entropy-based and t-SNE (Van der Maaten & Hinton, 2008). Through entropy regularization we encourage learned activations to approximately select one element of a family of predefined activations (with applications to efficiency). Whereas, with t-SNE regularization we promote that similar/dissimilar conditioning vectors yield similar/dissimilar learned activations, thereby instilling the geometry of the speaker embedding space in the conditioning mechanism (for enhanced conditioning quality).

### A.2.1 APPROXIMATELY ONE-HOT ASSIGNMENTS AND GEOMETRY

We explore mainly two regularization techniques to promote certain qualitative properties in the learned activations during training. The first technique encourages the learned activations to approximately select one element of $\{A_i : \mathbb{R} \to \mathbb{R}\}_{i=1}^{a}$, based on the conditioning vector. Different non-linear functions require different amount of computational resources, thus biasing the learned activations towards a single known form can improve efficiently, in comparison to their weighted combinations. The second technique guides training so that similar/dissimilar conditioning vectors yield similar/dissimilar learned activations, thereby preserving the geometry of the speaker embedding space in the conditioning mechanism. Considering a particular use-case of PSE, there might be some variation in the conditioning vectors (i.e., speech samples collected during enrolment). This turns into a challenge of selecting or aggregating these vectors to properly condition the network. The use of our second technique for regularization during training could reduce the impact of such variations in conditioning vectors at deployment phase by increasing the stability of the network against subtle variations in the conditioning vectors.

**Learning approximately one-hot assignments** Sparse assignments can be encouraged by adding known regularizers, such as entroypy or $\ell_1$ to the loss during training, scaled by a hyper-parameter $\alpha$, i.e., $\alpha\mathrm{H}(\boldsymbol{s})$ or $\alpha\|\boldsymbol{s}\|_1$. Alternatively, sparsity can be promoted by considering a temperature parameter $t \leq 1$. For example, a value lower than one makes the model more confident, whereas higher than one makes it less so. Thus sparse activation can be learned as: $\boldsymbol{s} := \mathrm{softmax}_{\mathrm{rowwise}}(t^{-1}(\boldsymbol{z} \times \boldsymbol{w} + \boldsymbol{b}))$.

**Approximately preserving conditioning vectors geometry** Learned activations are built based on conditioning vectors, and we may expect these functions to have a discriminative property for the conditioning vectors they are based on. In other words, clusters in $\boldsymbol{z}_j$ space would be correlated to clusters in the space of learnt conditioned activations weights $\boldsymbol{s}_j$ for similar/dissimilar conditioning vectors. However, strong correlation may not happen automatically due to: i) the dimension of the conditioning vectors is often much larger than the number of basic activation functions $a$, i.e., $d \gg a$, ii) the manner in which $\boldsymbol{z}_j$ is used to produce $\boldsymbol{s}_j$ may be affected by the fact that $\mathrm{softmax}$ is invariant to translation, i.e., $\mathrm{softmax}(\boldsymbol{v} + c) = \mathrm{softmax}(\boldsymbol{v})$, and iii) the similarity/dissimilarity of the basic activations $\{A_i : \mathbb{R} \to \mathbb{R}\}_{i=1}^a$. High correlation can be promoted though by geometry aware high- to low-dimensional embeddings like t-SNE (Van der Maaten & Hinton, 2008). Although typically used for data visualization, we instead leverage the t-SNE objective function as a regularizer in our loss function. Specifically, we take t-SNE probabilities $\boldsymbol{p}, \boldsymbol{q} \in [0,1]^{b \times b}$ defined by $\boldsymbol{p}_{i,i} := \boldsymbol{q}_{i,i} := 0$ and for $i \neq j$:

$$p_{j|i} := \frac{e^{-\|\boldsymbol{z}_i - \boldsymbol{z}_j\|^2/2\sigma^2}}{\sum_{k \neq i} e^{-\|\boldsymbol{z}_i - \boldsymbol{z}_k\|^2/2\sigma^2}}, \quad \boldsymbol{p}_{i,j} := \frac{p_{j|i} + p_{i|j}}{2b}, \quad \boldsymbol{q}_{i,j} := \frac{(1 + \|\boldsymbol{s}_i - \boldsymbol{s}_j\|^2)^{-1}}{\sum_k \sum_{l \neq k}(1 + \|\boldsymbol{s}_k - \boldsymbol{s}_l\|^2)^{-1}},$$

and add $\beta\mathrm{KL}\left(\boldsymbol{p} \parallel \boldsymbol{q}\right)$ to our loss function, where $\beta$ is a hyper-parameter and KL denotes the Kullback–Leibler divergence. The minimization of the KL regularizer may be useful in applications, where conditioning vectors are naturally clustered such as in audio applications involving speaker embedding vectors, such as PSE, TTS, and speaker-dependent ASR. Also in applications with discrete latent vectors (Oord et al., 2017), where the preservation of the geometry of the conditioning vectors in the learned activations may offer superior results or better insights into the modelling internals.

### A.2.2   IMPACT OF REGULARIZERS

We now investigate the impact of using regularizers for training PSE models. We trained all models with two regularizes (c.f §A.2), which are used individually to analyze their unique benefits (as discussed earlier). Tables 5–6 present the performance of all models by using each regularizer with models trained on two datasets. To show the effectiveness of the regularizers, we compare these results against those in Tables 1–2 for models learned without regularizers.

The results clearly demonstrate the improvement in performance of RNN and TDS models when trained with either of the regularizers. The performance gain is consistent across both datasets, but there is a trade off with the impact on clean speech. Moreover, we observe that entropy regularizer is outperformed by t-SNE regularizer. This could be due to the fact that t-SNE regularizer is less restrictive allowing a wider range of combinations of the basic activations. Whereas, entropy regularizer promotes the selection of only one of the basic activations.

Interestingly, the TDS-RNN models, which performed best without regularizers, get worse performance with regularizers. This could be due to limited number of filter layers (i.e., only 2, compared with 8 and 42 for RNN and TDS models respectively) present in the model architecture. Given that LA was given only 11 basic activation to learn from, there could be merely 121 (i.e., $11^2$) unique separations learned by LA. This makes it difficult for these models to successfully differentiate between all unique conditioning vectors which might contain way more separations.

Note, we considered as hyper-parameters only the values in $\alpha, \beta \in \{1\mathrm{e}{-5}, 1\mathrm{e}{-4}, 1\mathrm{e}{-3}, 1\mathrm{e}{-2}\}$, $\sigma \in \{1\}$ and used the same learning rates as for training models without regularizers. Due to limited computational budget, we could not perform an exhaustive search on these hyper-parameters.

### A.2.3   CORRELATION ANALYSIS

To understand the manner in which the models leverage the conditioning vectors together with inputs to predict outputs, we perform a correlation analysis between distance matrices of softmax values (of LA) and of conditioning vectors. For a random selection of embedding vectors $\{\boldsymbol{z}_j\}_{j=1}^{100}$, we computed the weights $\boldsymbol{s}_j^{(l)}$ with $j \in \{1, \ldots, 100\}$ of the $l$ LAs of a model. Now, we compute the distance

Table 5: SDRi and WER results on LibriSpeech (English) when using learned activations (LA) trained with entropy ($\alpha$) and t-SNE ($\{\beta, \sigma\}$) regularizers.

| Model | Regularizers | | SDRi | | WER | | |
|---|---|---|---|---|---|---|---|
| | Entropy | t-SNE | Babble | Ambient | Clean | Babble | Ambient |
| RNN | 1e−4 | {0, 1} | 8.53 | 11.90 | 8.5 | 24.5 | 15.4 |
| TDS | 1e−3 | {0, 1} | 7.27 | 11.55 | 8.3 | 33.0 | 16.0 |
| TDS–RNN | 1e−5 | {0, 1} | 7.95 | 12.14 | 8.2 | 28.8 | 15.2 |
| RNN | 0 | {1e−4, 1} | 8.46 | 12.08 | 8.4 | 24.3 | 15.0 |
| TDS | 0 | {1e−5, 1} | 7.42 | 11.42 | 8.5 | 33.0 | 16.1 |
| TDS–RNN | 0 | {1e−5, 1} | 8.51 | 12.50 | 8.2 | 25.4 | 14.9 |

Table 6: SDRi and WER results on VoxForge (Spanish) when using learned activations (LA) trained with entropy ($\alpha$) and t-SNE ($\{\beta, \sigma\}$) regularizers.

| Model | Regularizers | | SDRi | | WER | | |
|---|---|---|---|---|---|---|---|
| | Entropy | t-SNE | Babble | Ambient | Clean | Babble | Ambient |
| RNN | 1e−5 | {0, 1} | 7.08 | 9.74 | 1.6 | 18.6 | 5.9 |
| TDS | 1e−5 | {0, 1} | 5.85 | 8.61 | 2.1 | 27.0 | 7.1 |
| TDS–RNN | 1e−5 | {0, 1} | 5.51 | 9.34 | 1.3 | 33.3 | 6.8 |
| RNN | 0 | {1e−5, 1} | 7.36 | 10.12 | 2.6 | 16.9 | 5.9 |
| TDS | 0 | {1e−2, 1} | 5.68 | 8.62 | 1.8 | 29.0 | 7.1 |
| TDS–RNN | 0 | {1e−5, 1} | 6.33 | 9.92 | 1.4 | 25.2 | 6.5 |

matrices for $z_j$ and $s_j^{(l)}$, which we refer to as $Z$ and $S^{(l)}$. Here, the distance matrix is computed by taking the pair-wise cosine distances. To get insights about how LAs perform conditioning, we compute the Spearman correlation between $\text{vec}(Z)$ and $\text{vec}(S^{(l)})$. This correlation is computed for all LAs (i.e., 8, 42 and 2 for RNN, TDS, TDS–RNN models). This allows to better measure the impact of the conditioning vector across the depth (i.e., position of the LA) in the network.

In Figure 6 (bottom rows of each sub-figure) we present the correlation values for each LA in all three models. We observe there is a large variability in correlation values as the number of layers increases. We do not see very strong correlation with a specific LA, which might be due to the fact that conditioning is performed at different stages.

In addition, we also present (shown in top rows of each sub-figure of Figure 6) the distance matrices $S^{(l)}$ (for $l$ which has the highest correlation), as well as $Z$ (in the last column). The overall structure of $Z$ is approximately block diagonal with three blocks, is not well preserved when learning without regularizers given the blocks collapse to two, but is well preserved when learning with regularizers. This is especially significant given that conditioning vectors are high dimensional (in this example 256) but the weight vectors in LAs are low dimensional (in this case 11). We also observe that the models with less number of LAs preserve much similar structure between $S^{(l)}$ and $Z$. Given the networks have similar performance, this indicates that as the number of LAs decrease, the networks enforce higher degree of conditioning per LA.

### A.3    IMPACT OF ENROLLMENT DATA ON AMBIENT NOISE

Tables 5 and 6 show the performance of various PSE models based on two seconds of enrollment data in terms of babble and ambient noise. Although babble noise, i.e., other people talking, cannot be expected to be suppressed without knowledge of the owner's voice profile, clean and ambient noise on the other hand does not in principle require such enrollment data. To investigate the performance of the various PSE models in the absence of enrolment data we have replaced it with vectors of random, zeros and ones, and measured the quality of PSE on ambient noise in Tables 7–8, which demonstrate that all architectures work well (compared to concatenation based conditioning approach) even without enrolment data for clean and ambient noise when using ones instead of conditioning vectors. We also see that for the best architecture (TDS–RNN) that is the case not just with replacing by ones but also by random and zeros.

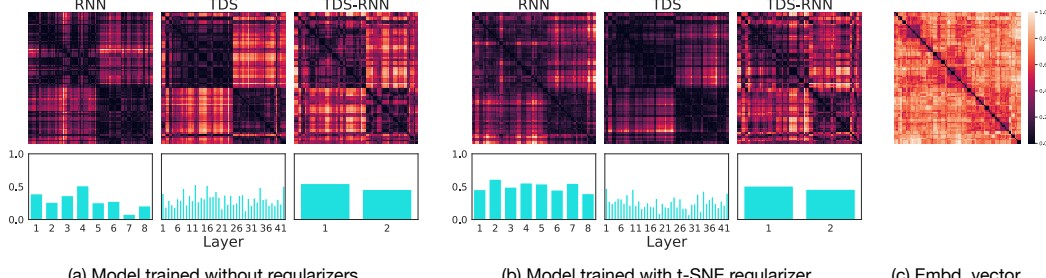

(a) Model trained without regularizers     (b) Model trained with t-SNE regularizer     (c) Embd. vector

Figure 6: Correlation analysis of conditioning vectors with softmax values of LA for models trained on Librispeech. In sub-figure (a) and (b), the bottom row shows the overall correlation of all LAs in model with conditioning vectors, and the top row depicts the distance matrix corresponding to LA with the highest overall correlation for each model as well as for conditioning vectors (fig c).

Table 7: Results for models trained on Librispeech (English) data with proposed LA and baseline concatenation and evaluated with random, zeros and ones as speaker embedding vectors.

| Model | LA | SE | SDRi | | WER | | |
|---|---|---|---|---|---|---|---|
| | | | Babble | Ambient | Clean | Babble | Ambient |
| Identity | - | - | - | - | 8.2 | 89.4 | 17.3 |
| RNN | LA | Random | 3.08 | 9.24 | 10.6 | 48.8 | 15.7 |
| RNN | LA | Zeros | 0.47 | 8.41 | 25.9 | 60.2 | 21.8 |
| RNN | LA | Ones | 4.61 | 11.64 | 9.7 | 44.4 | 15.6 |
| RNN | Concat. | Random | 1.17 | 6.32 | 28.2 | 55.2 | 20.0 |
| RNN | Concat. | Zeros | 3.70 | 10.76 | 20.2 | 43.8 | 17.3 |
| RNN | Concat. | Ones | 2.20 | 11.44 | 10.8 | 51.1 | 15.8 |
| TDS | LA | Random | -3.48 | 4.36 | 11.3 | 79.6 | 19.7 |
| TDS | LA | Zeros | 2.65 | 4.58 | 14.4 | 85.5 | 25.5 |
| TDS | LA | Ones | -0.59 | 10.59 | 8.4 | 74.0 | 16.1 |
| TDS | Concat. | Random | -2.90 | 5.63 | 32.7 | 74.6 | 24.1 |
| TDS | Concat. | Zeros | -0.68 | 8.38 | 30.9 | 68.5 | 21.7 |
| TDS | Concat. | Ones | -1.13 | 9.06 | 24.8 | 67.4 | 19.5 |
| TDS–RNN | LA | Random | 1.99 | 12.23 | 14.0 | 52.2 | 14.4 |
| TDS–RNN | LA | Zeros | 1.91 | 12.19 | 17.0 | 52.8 | 14.5 |
| TDS–RNN | LA | Ones | 3.90 | 12.33 | 9.5 | 44.8 | 14.4 |
| TDS–RNN | Concat. | Random | 1.79 | 7.42 | 58.2 | 46.1 | 21.0 |
| TDS–RNN | Concat. | Zeros | 7.33 | 11.84 | 25.4 | 25.6 | 15.2 |
| TDS–RNN | Concat. | Ones | 3.66 | 11.75 | 23.1 | 39.8 | 15.4 |

Table 8: Results for models trained on VoxForge (Spanish) data with proposed LA and baseline concatenation and evaluated with random, zeros and ones as speaker embedding vectors.

| Model | LA | SE | SDRi | | WER | | |
| --- | --- | --- | --- | --- | --- | --- | --- |
| | | | Babble | Ambient | Clean | Babble | Ambient |
| Identity | - | - | - | - | 1.2 | 89.1 | 9.2 |
| RNN | LA | Random | -3.23 | -3.00 | 76.6 | 77.8 | 52.7 |
| RNN | LA | Zeros | -1.60 | 2.45 | 48.7 | 64.9 | 33.3 |
| RNN | LA | Ones | -3.53 | 1.37 | 61.5 | 69.7 | 37.7 |
| RNN | Concat. | Random | -1.77 | -0.44 | 56.9 | 73.6 | 41.0 |
| RNN | Concat. | Zeros | -1.26 | 2.34 | 42.9 | 63.4 | 30.7 |
| RNN | Concat. | Ones | -2.09 | 1.02 | 57.9 | 67.8 | 39.2 |
| TDS | LA | Random | -3.91 | -3.21 | 49.8 | 85.8 | 40.3 |
| TDS | LA | Zeros | -3.91 | -3.21 | 49.8 | 85.8 | 40.3 |
| TDS | LA | Ones | -3.84 | 0.18 | 33.4 | 78.7 | 30.4 |
| TDS | Concat. | Random | -3.80 | -2.48 | 34.1 | 84.0 | 33.6 |
| TDS | Concat. | Zeros | -4.05 | -1.06 | 39.9 | 80.2 | 39.5 |
| TDS | Concat. | Ones | -4.06 | -1.51 | 47.2 | 78.1 | 42.2 |
| TDS–RNN | LA | Random | -1.07 | 8.37 | 13.0 | 56.8 | 10.3 |
| TDS–RNN | LA | Zeros | -1.13 | 8.26 | 14.0 | 56.9 | 10.9 |
| TDS–RNN | LA | Ones | -0.85 | 8.58 | 10.7 | 56.4 | 9.4 |
| TDS–RNN | Concat. | Random | -0.10 | 0.88 | 86.6 | 56.3 | 39.9 |
| TDS–RNN | Concat. | Zeros | 1.59 | 6.09 | 56.2 | 42.7 | 19.9 |
| TDS–RNN | Concat. | Ones | 0.47 | 3.75 | 66.3 | 51.1 | 31.7 |

