# OpenReview forum: "Conditioning Sequence-to-sequence Networks with Learned Activations"
_ICLR.cc/2022/Conference — ICLR 2022 Poster_

### Official Review · Reviewer_dsMi · 2021-11-02

**Correctness:** 4
**Technical Novelty And Significance:** 3
**Empirical Novelty And Significance:** 2
**Recommendation:** 6
**Confidence:** 4

**Main Review:**

Strengths:
- Extensive experiments on two application areas (speech enhancement and automatic speech recognition).
- The paper also tried to provide quantitative results by plotting learned activations for different speakers.
- The proposed model is comparably smaller than the concatenation based approaches. The method might be suitable for on-device applications.

Weaknesses:
- As there are many experimental results, the paper may occasionally become harder to read towards the end.
- Some details are not well-described. For example, in Tables 1 and 2, what does #Cond. mean?
- Even though the model size is smaller, it comes with additional on-the-fly computation of several non-linearities of hidden activations, which may cause other concerns for on-device applications.
- FiLM approach should be briefly mentioned in the text before reporting the results with this method.
- HPO acronym should be defined. (Page 6)
- How exactly does the study perform fine-tuning of the speaker embedding model? Details are missing. (Page 8)
- For PASR, does the study fine-tune the speaker embedding model jointly with the ASR model or are they learned separately?

**Summary Of The Paper:**

This paper introduces a method called learned activations for personalized speech enhancement and personalized automatic speech recognition. The learned activations are obtained by a weighted sum of nonlinear activations of hidden layer outputs and the weights are computed from softmax of the projected speaker embeddings such as x-vectors. The main benefit of the model as compared to concatenation or modulation based approaches is that Learned Activations result in smaller number of learnable parameters and hence a smaller model size. Experimental results show that in terms of SDR improvement, the proposed method achieves comparable performance to the baseline models with a smaller model size. In the case of ASR, the proposed approach provides WER reduction as compared to an unadapted model.


**Summary Of The Review:**

The paper provides a personalization method by conditioning non-linear activations based on the speaker embeddings. There are extensive experiments. One concern is that the paper may need some additional clarification on details for better reproducibility.

---

> ### Author Response · Authors · 2021-11-16
> **Improved presentation**
>
> 1) **Improve readability in discussion**:
>
> Thanks very much for this comment, and based on all the comments we have tried to improve the readability of the paper. In particular:
>
> - Updated the notations used in the equations to have direct comparisons with the figures.
> - Added clarification text to describe the main idea better.
> - Updated captions with definition to describe the illustrations better.
> - Improved the labeling texts in the figures.
> - Rearranged text and added more clarity to the tables.
>
> 2) **Clarify table headers notation**:
>
> We expanded Section 4.2 to clarify how many conditioning layers, i.e., \#Cond. are used per model, and expanded the caption of Tables 1 and 2 to also clarify further. The new text reads:
> "Furthermore, we consider a different number of conditioning layers (\#Cond.) for each model family. In particular, for RNN 8, for TDS 21 and for TDS--RNN 2 conditioning layers leading to 8, 42 and 2 nodes in the models being influenced by the conditioning information (c.f., Figure 1). We have done so to investigate also how the number of conditioning layers influences the personalization effects from the conditioning vectors (c.f., Appendix A.1).''
>
> 3) **Describe FiLM modulation before reporting results**:
>
> We expanded Section 4.2 to clarify the selection of the FiLM layer as a representative for modulation approaches. The new text reads:
> "As a representative of light weight-modulation approaches, we consider the Feature-wise Linear Modulation (FiLM) (Perez et al., 2018) mechanism throughout."
>
> 4) **Define HPO**:
>
> Apologies for this lapse. We have defined the term Hyper-parameter optimization (HPO) in the text.
>
> 5) **Clarify speaker embedding training for PASR**:
>
> There are two sets of experiments for PASR.
> The first, referred to in the text as "train selected parameters"/"train LA", considers as trainable parameters only those of the speaker embedding models as well as the weights as biases of the LAs.
> The second, referred to in the text as "train all parameters"/"train all", additionally consider also the trainable parameters from the ASR model.
> In both sets of experiments, the loss function is the same, namely the CTC loss. We have adding this clarity in the text.

---

> ### Author Response · Authors · 2021-11-16
> **FLOPS and latency measurements**
>
> We have added FLOPS and latency measurements in the paper and below, which clarifies that there isn't any significant impact of additional on-the-fly computation of several non-linearities of hidden activations:
>
> | Model | Type | FLOPs (M) | Latency CPU (ms) | Latency GPU (ms) |
> |----------------|:-------------:|:------------------:|--------------------------:|--------------------------:|
> | RNN            |       LA      |       1254.85      |     462.997 ($\pm$14.719) |      39.156 ($\pm$01.339) |
> | RNN            |      FiLM     |       1254.85      |     465.785 ($\pm$14.612) |      35.769 ($\pm$01.344) |
> | RNN            |    Concat.    |       1802.60      |     537.766 ($\pm$11.403) |      46.716 ($\pm$00.903) |
> | TDS            |       LA      |       1507.43      |     127.420 ($\pm$02.996) |      78.998 ($\pm$01.963) |
> | TDS            |      FiLM     |       1507.43      |      95.535 ($\pm$04.861) |      70.265 ($\pm$01.975) |
> | TDS            |    Concat.    |       2906.46      |     123.580 ($\pm$03.440) |      84.614 ($\pm$02.479) |
> | TDS--RNN       |       LA      |       4406.60      |     380.014 ($\pm$16.604) |     233.748 ($\pm$05.315) |
> | TDS--RNN       |      FiLM     |       4406.60      |     387.325 ($\pm$22.101) |     238.787 ($\pm$11.882) |
> | TDS--RNN       |    Concat.    |       4543.92      |     394.809 ($\pm$20.247) |     237.437 ($\pm$06.928) |

---

### Official Review · Reviewer_QdFv · 2021-11-02

**Correctness:** 3
**Technical Novelty And Significance:** 3
**Empirical Novelty And Significance:** 3
**Recommendation:** 6
**Confidence:** 3

**Main Review:**

The experimental results indicate that the proposed method of training the RNN / CNN based systems for speech enhancement and ASR leads to reduction in number of parameters. The authors argue that this is an important advantage for low-resource computing. I have several questions on this. I have reviewed this paper before, and I see that some of my concerns from earlier, even though were partially adressed in the rebuttal, now seems not to be addressed in the current version of the manuscript.

The number of parameters to assess the computational advantages of a model is questionable in my opinion. I think a better measure for this would be to use the number of FLOPs in forward pass, or the memory usage with respect to input sequence length. Namely, this table was presented in the rebuttal:

Model 	LA 	FLOPs (M) 	Latency (ms)

RNN 	y 	1254.85 	528.543 (+-36.392)

RNN 	n 	1802.60 	537.766 (+-11.403)

TDS 	y 	1507.43 	116.451 (+-06.992)

TDS 	n 	2906.46 	123.580 (+-03.440)

TDS--RNN 	y 	4406.60 	392.326 (+-15.582)

TDS--RNN 	n 	4543.92 	394.809 (+-20.247)

I think it would be good to include these results. Overall, we see that the improvement is not always clear, especially in terms of latency, when the variance is taken into consideration.

For the new experiment with ASR, and PASR, (part 2 of the experiments) the presentation is not super clear to me, and I am not sure if an improvement over alternative conditioning techniques are shown here.

**Summary Of The Paper:**

This paper proposes a way of conditioning information on neural networks. In literature a common way to condition a neural with an input would be to either concatenate the conditioning vector to the input vector, or inject it before several layers (modulation approach in Figure 1-b). In this paper they instead propose to pass the conditioning vector through weighted sum of output of neural-net activation functions. The claim is that this way of training the neural network leads to reduction in neural network parameters without sacrificing performance in speech enhancement / ASR tasks.

**Summary Of The Review:**

I was a reviewer for this paper for Neurips 2021. Even though I voted for acceptance that time, I now see that some of my concerns and concerns pointed out by other reviewers are not fully addressed. For instance I do not see a comparison with this paper: https://arxiv.org/pdf/1601.02828.pdf  . I also do not see a comparison in terms of # FLOPS and latency for which the improvement seems marginal from the results added to the Neurips rebuttal.

I therefore vote for marginal rejection.

---

> ### Author Response · Authors · 2021-11-16
> **Add modulation baselines for personalized ASR**
>
> For the sound enhancement experiments we took popular FiLM as the representative modulation technique and use it as the baseline. As part of this rebuttal, we have opted to do so also for the personalized ASR based on the reviewers suggestion. We believe that conditioning via FiLM, rather by https://arxiv.org/pdf/1601.02828.pdf, provides the reader a more holistic comparison between our proposed LA approach and existing conditioning techniques across two application domains, as opposed to having different modulation approaches for different application domains (namely personalized ASR and sound enhancement).
>
> Furthermore, for a fair comparison, as a similar mechanism to the bias offset in LAs, we altered FiLM's definition $A_\mathrm{elementwise}(G_{\eta}(U_{\mu}(z_j) \odot x_{j,\ldots,:} + V_{\nu}(z_j)))$ to $A_{elementwise}(G_{\eta}((1 + \alpha\cdot tanh(U_{\mu}(z_j))) \odot x_{j,\ldots,:} + \beta\cdot tanh(V_{\nu}(z_j))))$ with $\alpha$ and $\beta$ as trainable variables, so that at initialization FiLM is approximately the identity function for most conditioning vectors. We found that this modification produces better results (c.f., Section 5.2 in the latest version) in the case of personalized ASR.
> We present the new personalized ASR via FiLM results in Table 4 (Table 3 in previous version):
>
> |  Models |  ASR | ASR | ASR | ASR | PASR | PASR | PASR | PASR | PASR | PASR | PASR | PASR |
> |:----------------:|:----:|:----:|:-------:|:----:|:----:|:----:|:-------:|:----:|:----:|:----:|:-------:|:----:|
> |   Type  |  -  | - | - | - | FiLM | FiLM | FiLM | FiLM |  LA  | LA | LA | LA |
> | Encoder | Char | Char | Subw | Subw | Char | Char | Subw | Subw | Char | Char | Subw | Subw |
> | Decoder |   G  |  BS  |    G    |  BS  |   G  |  BS  |    G    |  BS  |   G  |  BS  |    G    |  BS  |
> |   WER   | 8.93 | 8.77 |   7.69  | 7.62 | 7.41 | 7.27 |   6.99  | 6.93 | 7.26 | 7.15 |   6.41  | 6.35 |
> |   CER   | 2.65 | 2.62 |   5.04  | 5.01 | 2.20 | 2.18 |   4.78  | 4.75 | 2.17 | 2.14 |   4.39  | 4.36 |

---

> ### Author Response · Authors · 2021-11-16
> **FLOPS and latency measurements**
>
> We were very fortunate and grateful to the four reviewers of our previous submission to NeurIPS, which provided us with many constructive comments and suggestions for improvement. During the discussion period we addressed a very significant portion of the comments, amounting over 30 initial replies and additional follow-up replies.
>
>
>
> In our opinion at the time, as highlighted by the program Chair's final recommendation, the reviewers were mostly interested in seeing another application area (our NeurIPS submission had the PSE part of the current manuscript, but didn't have the personalized ASR part of the current manuscript), rather than presenting detailed FLOPs and latency measurements.
>
> The main reasons for excluding the FLOPs and latency results, as presented during our NeuRIPS rebuttal, are as follows:
>
> - Although, the FLOP and latency results showed competitive performance, as can be seen in the table provided by the reviewer, they were computed without any implementation of custom kernels. Namely, there was no operation fusion and in particular memory for every basic activation was allocated and written to, i.e., our implementation was inefficient.
>
> - Given the short time between NeuRIPS notifcation and ICLR submission, and since it was our understanding that FLOPs/latency was not a critical point in the reviewers mind, we gave more priority in expanding to a different application domain. Therefore based on the suggestion of NeurIPS reviewers, we selected personalized ASR as the new domain and added new extensive experimental findings to the current manuscript.
>
> That said, we strongly agree with the reviewer that the FLOPs and Latency measurements would improve the quality of the paper and  therefore, we will add this to the current manuscript.
> Furthermore, we have implemented fused implementations in CPU as well as in GPU with C++ back-end for the conditionally learned activations, which we report in the updated version of the manuscript in new Table 3:
>
> | Model | Type | FLOPs (M) | Latency CPU (ms) | Latency GPU (ms) |
> |----------------|:-------------:|:------------------:|--------------------------:|--------------------------:|
> | RNN            |       LA      |       1254.85      |     462.997 ($\pm$14.719) |      39.156 ($\pm$01.339) |
> | RNN            |      FiLM     |       1254.85      |     465.785 ($\pm$14.612) |      35.769 ($\pm$01.344) |
> | RNN            |    Concat.    |       1802.60      |     537.766 ($\pm$11.403) |      46.716 ($\pm$00.903) |
> | TDS            |       LA      |       1507.43      |     127.420 ($\pm$02.996) |      78.998 ($\pm$01.963) |
> | TDS            |      FiLM     |       1507.43      |      95.535 ($\pm$04.861) |      70.265 ($\pm$01.975) |
> | TDS            |    Concat.    |       2906.46      |     123.580 ($\pm$03.440) |      84.614 ($\pm$02.479) |
> | TDS--RNN       |       LA      |       4406.60      |     380.014 ($\pm$16.604) |     233.748 ($\pm$05.315) |
> | TDS--RNN       |      FiLM     |       4406.60      |     387.325 ($\pm$22.101) |     238.787 ($\pm$11.882) |
> | TDS--RNN       |    Concat.    |       4543.92      |     394.809 ($\pm$20.247) |     237.437 ($\pm$06.928) |

---

### Official Review · Reviewer_gotN · 2021-11-08

**Correctness:** 3
**Technical Novelty And Significance:** 4
**Empirical Novelty And Significance:** 3
**Recommendation:** 6
**Confidence:** 3

**Main Review:**

See my comments above in the Summary for strengths and weaknesses. In an nutshell, I think the paper suffers from a lack of clarity that undersells what is otherwise a nice idea.

I suggest introducing the specifics of the core of the idea earlier on in the paper. There's a sense of anticipation that is a bit frustrating for me as a reader.

Some more specific comments follow.

Re: sentence ending in, "... and a family of `a` basic activations {Ai : R → R} a i=1 of which a particular realization could be the set of the most commonly used activations in deep learning (e.g. relu, sigmoid, tanh, and so on)": check the grammar, I believe this is not a complete sentence given what comes before the sub-clause I just quoted.

What is "softmax_rowwise"?

Re: First 3 equations in Section 3.1: What is the relationship between LA_elementwize(c | z_j) and LA(h_j | z_j)? (Numbering the equations would be helpful). What is c?

I did not quite follow the formalism. I am unable to clearly relate the equations in 3.1 with Figure 1c.

A number of references are given for the concatenation approach, which is good. But no references are given for the modulation approach (AFAICT), though this is referred to as a state-of-the-art approach. Here again, I am unable to follow the discussion linking the formalism to Figure 1b. I am guessing that the formalism is correct but the presentation could be substantially simplified and clarified.

Figures 2: "To take into account that basic activations may have different ranges, the plots show LAs minus average of basic activations, i.e., LA(c | zj )  [...] for c ∈ [−3, 3] and values of j corresponding to the selected users." I cannot relate LA(c | zj) to the label of the x-axis, which is h.

Figure  4. "To take into account that the non-personalized basic activations are relu or swish, the plots show LAs minus average of those two activations, i.e., LA(c | zj ) − (relu(z) + swish(z))/2 for c ∈ [−3, 3] and values of j corresponding to the selected users. " I suggest labeling the x-axis. Would that be z? How should we relate z_j to the z (without the j subscript) argument of relu() and swish()?

Section 4.1, Table 1: Define SDRi before using the term.

Section 4.2, "HPO": "Hyper-parameter optimization"?

Section 4.4, "STFT": define the term.

Section 5.4, "Greedy search": provide a reference? Later, in the results section, the authors state, "Furthermore, the gains can be seen to be more pronounced for greedy than beam search CTC decoders, reaching up to 40% relative improvement for some users which is a significant improvement in the domain of ASR," but greedy search doesn't represent the typical final ASR benchmark, so the greater relative gain than for beam search doesn't seem too significant -- but I am not sure what the authors mean by greedy search, so they should clarify this.

**Summary Of The Paper:**

This paper presents a very interesting idea (if I understand it correctly), which is to implement conditioning on e.g. a speaker embedding vector via a parametrization of the layer-to-layer activation functions of an overall deep neural net architecture. The motivation is to achieve the same effect as the well-known concatenation approach, but without requiring as many new parameters.

I found the formalism hard to follow, impeding my grasp of the fundamental proposal. However, eventually I got the idea. The results are promising, showing good preservation of e.g. WER quality on the unconditioned scenario, with good WER on the specialized scenario, and with fewer parameters than the concatenation method.

I think the name "learned activations" somewhat undersells the idea -- wouldn't it be better named as "learned activation _functions_"? That immediately would clarify the fundamental concept -- and highlight it's originality.

**Summary Of The Review:**

Good originality (AFAICT) and good results; somewhat unclear formalism and presentation of main concepts.

---

> ### Author Response · Authors · 2021-11-16
> **Improved presentation**
>
> 1) **Replace learned activations with learned activation functions**:
>
> We thank the reviewer for this excellent suggestion and accordingly we have updated the title and text of the manuscript.
>
> 2) **Correct grammar in sentence introducing set of basic activations**:
>
> We have simplified this sentence. In particular, the highlighted sentence now ends with: "... and $\{A_i:\mathbb{R}\rightarrow\mathbb{R}\}_{i=1}^a$ is an ordered family of $a$ basic activation functions, e.g., $\{\mathrm{relu}, \mathrm{sigmoid}, \mathrm{tanh}\}$".
>
> 3) **Define $\mathrm{softmax}_\mathrm{rowwise}$**:
>
> In section 3.1 of the manuscript we include $b$, i.e., the batch size dimension, in the equations. Thus, individual samples $\boldsymbol{z}_j\in\mathbb{R}^{1\times d}$ in $\boldsymbol{z}\in\mathbb{R}^{b\times d}$ correspond to embeddings extracted from diffrent audio utterances. Now, the weights $\boldsymbol{s}_j\in\mathbb{R}^{1\times a}$ for the basic activations are computed by applying the affine transformation, i.e., $\boldsymbol{s}_j = \boldsymbol{z}_j\boldsymbol{w} + \boldsymbol{b}$. The notation $\mathrm{softmax}_\mathrm{rowwise}$ was our attempt to emphasize this.
> We have clarified this in the text by writing "where $\mathrm{softmax}_\mathrm{rowwise}$ denotes the matrix-matrix operation with entries defined rowwise via $\mathrm{softmax}_\mathrm{rowwise}(\boldsymbol{m})_j:=\mathrm{softmax}(\boldsymbol{m}_j)$".
>
> 4) **Clarify LA with and without elementwise subscript**:
>
> We are sorry for the confusion.
> Given $h_{j, \ldots}$ is a tensor, by the subscript $LA_{elementwise}$ we meant to say that $LA(\cdot | z_j)$ should be applied to each element of $h_{j, \ldots}$.
> We have clarified this in the text by writing "and $LA_{elementwise}(h_{j, \ldots} |  z_j)$ denotes the result of applying $LA(\cdot | z_j)$ elementwise to each entry of $h_{j, \ldots}$".
> The variable $c$ corresponds to one of the elements of $h_{j, \ldots}$, but a much better notation would be, as the reviewer implicitly points out, to change the lower case $c$ with lower case $h$. Thank you for bringing these points to our attention and we have updated the equations as:
> \begin{align*}
>     s &:= softmax_{rowwise}(z\times w + b), \newline
>     h\in\mathbb{R}\mapsto LA(h | z_j) &:= \sum_{i=1}^as_{j, i} \cdot A_i(h), \newline
>     y_{j, \ldots} &:= LA_{elementwise}(h_{j, \ldots} |  z_j),
> \end{align*}
>
> 5) **Clearly relate the equations in 3.1 with Figure 1c**:
>
> Thanks very much for the comment, we have now replaced lower case $c$ with lower case $h$ to clarify the correspondence between equations in 3.1 and Figure~1.c. Especially, the modification allows for a direct correspondence on the roles of $z$, $h$ and $y$ between the formulas and the diagram.
>
> 6) **No references are given for the modulation approach**:
>
> Related work paragraph on "conditioning via normalization layers" presents the relevant reference (Perez el ta., 2018), but as the reviewer correctly states this is not mentioned to in the text. We have fixed this by adding a reference to it in Section 4.2.
>
> 7) **In Figure 2 relate LA, c and h**:
>
> Our apologies for this. In the caption $c$ should be h. The confusion is coming from the notation used in Figure 1 (in which we use $h$), and the formulas in Section 3.1 (in which we use $h$ and $c$).
> To address this issue we updated the following:
> (i) in the equations in Section 3.1 we replaced $c$ with $h$ to match what is used in Figure 1,
> (ii) in the captions of Figures 2 and 4 we replaced $c$ with $h$, e.g., in Figure 2 the caption now includes the updated formula $LA(h \,| \,z_j) - a^{-1}\sum_{i=1}^aA_i(h)$ for $h \in [-3, 3]$.
> Thank you for bringing this to our attention.
>
> 8) **In Figure 4 label x-axis, clarify $z$, relate $z_j$ with $z$, and arguments to the basic activations**:
>
> Thank you for pointing this out. There was a typo in the formula. In particular, $c$ should be replaced with $h$ (to better agree with the notation used in Figure 1) and $z$ should be replaced with $h$ like so: $LA(h \,| \,z_j) - (relu(h) + swish(h)) / 2$ for $h \in [-3, 3]$.
> We have also updated the x-labels in Figure-4 as $h$.
>
> 9) **Section 4.1, Table 1: Define SDRi before using the term**:
>
> We define the term SDRi in Section 5, which is before Table 1 is referred to in the text, but we agree this is not enough given that the table floated to an earlier page in the final submission efforts. We have therefore added the definitions also in the captions of the tables.
>
> 10) **Section 4.2, Define "HPO"**:
>
> Apologies for this lapse. We have now defined the term Hyper-parameter optimization in the text.
>
> 11) **Section 4.4, "STFT": define the term**:
>
> Apologies for this lapse. We have now defined the term Short Time Fourier Transform in the text.
>
> 12) **Section 5.4, "Provide reference for Greedy search decoder"**:
>
> By CTC greedy decoder we mean CTC beam search decoder with tops paths and beam width set to one. We have clarified this in the text.

---

> ### Author Response · Authors · 2021-11-16
> **Focus ASR/PASR discussion on beam search rather than greedy CTC decoders results**
>
> In Figure 5, the bottom row shows ASR and personalized ASR results when training everything, i.e., all parameters in the models. In particular, we can see from the first column (for CTC greedy decoder) and third column (for CTC beam search decoder) in Figure 5 bottom row, that the relative improvement for some users reaches 40\% for CTC greedy decoder and 35\% for CTC beam search decoder, which we believe are both significant improvements.
> Greedy search in particular plays an important role in on-device deployments for improved latency.
> We have added these clarifications to update the text.

---

### Author Response · Authors · 2021-11-16
**Summary**

We would like to sincerely thank all the reviewers for their time, effort and constructive comments on our manuscript. We addressed all of the reviewers comments.
In particular, the two major improvements have been:
Firstly, we have added another baseline namely FiLM modulation for personalized ASR.
Secondly, another significant improvement based on the suggestions from the reviewers has been to present comprehensive latency measurements based on efficient implementations of conditionally learned activations. We have uploaded updated versions of the main manuscript as well as the supplementary material.

- Update (2021/11/18): Updated FiLM personalized ASR WER/CER with latest results (now Table 4).

---

### Decision · Program_Chairs · 2022-01-20

**Decision:**

Accept (Poster)

**Comment:**

The authors propose a novel method for conditioning deep neural network. They replace the activation function with a linear combination of activation functions (e.g., ReLu). The weights for the activation functions are dynamically computed from the input during inference and training. The approach is evaluated on standard public tasks and shows improvement over well-established alternatives.

Pros
+ A simple novel method for condition that is widely applicable
+ Adequate empirical evaluations to demonstrate it's effectiveness

Cons
- No major weakness

The reviewers provided several feedback. The authors incorporated the suggestions and clarified residual concerns. The revised version of the paper has improved the readability and utility substantially.